# Overcoming false-positive gene-category enrichment in the analysis of spatially resolved transcriptomic brain atlas data

Ben D. Fulcher [1,2] ✉, Aurina Arnatkeviciute[2] & Alex Fornito [2]

Transcriptomic atlases have improved our understanding of the correlations between gene-expression patterns and spatially varying properties of brain structure and function. Gene-category enrichment analysis (GCEA) is a common method to identify functional gene categories that drive these associations, using gene-to-category annotation systems like the Gene Ontology (GO). Here, we show that applying standard GCEA methodology to spatial transcriptomic data is affected by substantial false-positive bias, with GO categories displaying an over 500-fold average inflation of false-positive associations with random neural phenotypes in mouse and human. The estimated false-positive rate of a GO category is associated with its rate of being reported as significantly enriched in the literature, suggesting that published reports are affected by this false-positive bias. We show that within-category gene–gene coexpression and spatial autocorrelation are key drivers of the false-positive bias and introduce flexible ensemble-based null models that can account for these effects, made available as a software toolbox.

[1] School of Physics, The University of Sydney, Camperdown, NSW, Australia. [2] The Turner Institute for Brain and Mental Health, School of Psychological Sciences and Monash Biomedical Imaging, Monash University, Clayton, VIC, Australia. ✉email: ben.fulcher@sydney.edu.au

The brain's multi-scale organization spans at least five orders of magnitude in space and time[1,2]. Understanding how these distinct scales relate to each other has proven challenging, largely because typical assays face a trade-off between spatial resolution and anatomical coverage. For example, in vivo brain-imaging techniques like magnetic resonance imaging (MRI) afford an unparalleled capacity to measure diverse aspects of structure and function across the entire brain, but have a limited spatial resolution that rarely surpasses 1 mm[3]. By contrast, invasive physiological recordings enable the measurement of neural structure and dynamics with cellular resolution, but traditionally only within small subregions of the brain. In recent years, our capacity to understand variations in molecular function across the entire brain has been greatly enhanced by industrialized, high-throughput transcriptome profiling, which has yielded genome-wide quantification of expression levels across the entire brain[3,4]. Two of the most influential genome-wide resources are the Allen Human Brain Atlas (AHBA), which encompasses microarray measurements of expression in >20,000 genes across 3702 tissue samples from six postmortem brains[4], and the Allen Mouse Brain Atlas (AMBA), which comprises in situ hybridization measures of expression for >17,000 genes at cellular resolution[3].

The availability of these genome-wide expression atlases has generated new opportunities to bridge spatial scales and uncover the microscale molecular correlates of macroscale brain organization. This correspondence has commonly been assessed by comparing regional variations in molecular function (from gene-expression maps) to independently measured macroscale structural and functional properties, e.g., from imaging techniques like MRI[5,6]. Prior work has characterized a relationship between gene expression and structural connectivity in *Caenorhabditis elegans*[7–10], mouse[11–18], and human[19,20]. Human research has further characterized links between gene-expression patterns and correlations of neural dynamics (functional connectivity) estimated from functional MRI (fMRI)[21–23] and electrocorticography[24]; brain morphometry and microstructure[25–30]; neurotransmitter receptor densities[31]; and disease-related changes in brain structure and function[32–35].

In these analyses, genes are typically scored according to the correspondence between their spatial expression map and anatomical variations of the independently measured phenotype, most commonly quantified as a Pearson correlation across a set of spatial locations. Instead of performing inference on thousands of genes directly, statistical tests can be performed at the level of groups of functionally annotated sets of genes, such as those involved in different types of biological processes. This process of gene category enrichment analysis (GCEA) investigates which categories are preferentially related to a spatial brain phenotype (SBP) of interest, reducing the genome-wide multiple comparison burden and facilitating biological interpretation. GCEA uses a statistical hypothesis-testing framework to assess which categories of genes are most strongly related to a given phenotype, leveraging annotations of genes to categories from open ontologies like the Gene Ontology (GO)[36] and KEGG[37]. GCEA has been applied extensively across species, scales, and diverse aspects of brain structure and function to gain insight into the biological processes that are mediated by genes with similar spatial expression profiles.

In the brain-imaging literature, a wide variety of publicly available tools have been used to perform GCEA[38–47]. While software packages make enrichment analysis easy to run, there are numerous challenges to correctly interpreting the results of GCEA. This requires careful consideration of (i) the reliability of gene annotations (including the rate of false-positive annotations[48]); (ii) the reproducibility of results (different software packages use different methods to perform enrichment with respect to the GO hierarchy; both annotations and GO terms are updated daily[49]); and (iii) the statistical inference procedure (the design of multiple-hypothesis testing across many non-independent and often hierarchically structured tests[48,50]). Methodological variability can lead to vast discrepancies in results obtained using different software packages. For example, one test (using identical inputs to different software packages) revealed a variation in $p$ values spanning several orders of magnitude for some GO categories[48]. Methodological developments of GCEA are ongoing[51,52].

Applications of GCEA, from the interpretation of genome-wide association studies[53], to case–control comparisons of microarray data, are subject to the general statistical challenges outlined above. But growing applications of GCEA to spatial transcriptional data—at the whole-brain as well as microscopic scale[54,55]—are associated with unique challenges due to the data's spatial embedding. These challenges have not yet been fully characterized or explored. First, spatial embedding introduces coexpression between genes with similar expression patterns[4,56,57], which results in gene–gene dependencies that form a generic characteristic of the expression dataset. Second, transcriptional maps are strongly spatially autocorrelated, such that nearby anatomical regions have more similar patterns of gene expression than distant regions, as has been observed in head neurons of *C. elegans*[10], the mouse brain[12,18], and human cortex[5,6,56,58–60]. The neural phenotypes that are matched to gene-expression atlas data are also commonly spatially autocorrelated[10,18,61–65]. Two spatial maps with similar spatial autocorrelation structure have a greater chance of exhibiting a high correlation to each other than two random spatial maps. Issues related to spatial autocorrelation of brain data have been highlighted in other contexts, with researchers developing methods to better estimate null distributions in the presence of spatial autocorrelation, e.g., using spatial permutation methods like spatial-lag models[30,66] and spin tests[67] to test against an ensemble of surrogate spatial maps, or by removing the effect of physical distance through regression[5,12,14,16–18,24,68]. Despite the growing acknowledgment of these issues, the precise impact of gene–gene coexpression and spatial autocorrelation of expression profiles on GCEA results has not been systematically characterized.

In this work, we evaluate the statistical biases involved in applying standard GCEA methodologies to spatially embedded transcriptomic datasets. We focus on whole-brain analyses here, but note that the same principles apply to GCEA analyses on any scale. We demonstrate that the rate at which a GO category is judged as significantly correlated to a random phenotype is far higher than statistical expectation, exceeding 20% for some GO categories. Analyzing a survey of the literature, we find a progressive increase in the rate of GO categories being reported as significant with their false-positive rate under random phenotypes, compatible with the reporting of false-positive bias. We show that these biases can be overcome using new ensemble-based null models that assess statistical significance relative to ensembles of randomized phenotypes (rather than constructing nulls by randomizing genes). Using case studies applying GCEA to a range of structural connectomic and cell density phenotypes in mouse and human brains, we show that highly significant categories under conventional GCEA are often consistent with ensembles of randomized phenotypes. Our ensemble-based approach to GCEA overcomes biases in investigating the transcriptomic correlates of spatially varying neural phenotypes, and is made available as a software toolbox[69].

## Results

We first describe a typical GCEA pipeline applied to a SBP of interest, depicted in Fig. 1A. The SBP is a spatial map of some measurement (e.g., taken across brain areas), such as gray matter

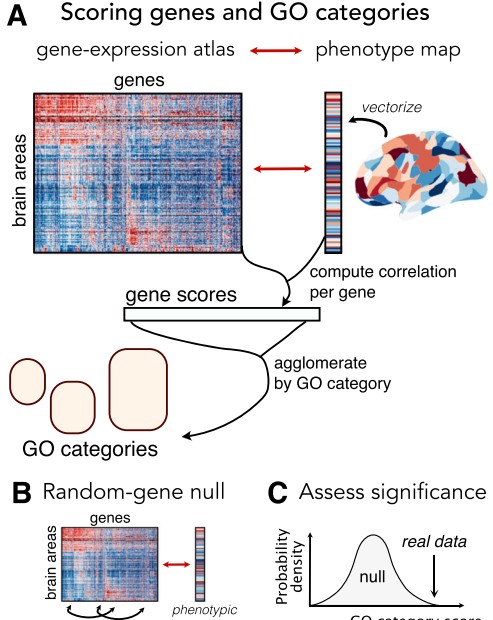

**A** Scoring genes and GO categories

gene-expression atlas ⟷ phenotype map

**Fig. 1 Pipeline for applying gene category enrichment analysis (GCEA) to brain-wide expression atlas data. A** Given a phenotype map, we first compute the spatial correlation coefficient between that map and each gene. These gene scores are then agglomerated at the level of categories using an annotation system like the Gene Ontology (GO). For continuous scores, agglomeration is typically performed as the mean score of genes annotated to a category. **B** Statistical significance of a GO category is assessed relative to the random-gene null, which estimates a null distribution for each GO category by annotating genes to GO categories at random. **C** For every GO category, a p value is estimated using a permutation test, by comparing the GO category score obtained from the real data to the null distribution.

volume, functional connectivity strength, or case–control differences in some property. Each gene in the atlas is scored using a measure of similarity between its expression pattern and the SBP (e.g., a correlation coefficient), yielding high scores for genes with expression maps that closely resemble the SBP. GCEA then tests whether high-scoring genes are concentrated in particular types of GO categories by comparing the GO category scores obtained from the data to those obtained from a null model. GCEA algorithms typically use a "random-gene null" to assess whether a GO category's score is higher than would be expected if genes were annotated to GO categories randomly[48]. This is similar to shuffling gene identities, as depicted in Fig. 1B, C. This general procedure captures the essence of applying GCEA to expression atlas datasets. We note that some pipelines for performing GCEA have been proposed that randomize phenotypes rather than genes[70,71], but to our knowledge they have not been applied in the setting of spatially embedded transcriptomics. We also note that, while here we focus on these types of spatial comparisons, some studies focus instead on pairwise phenotypes, such as the presence of a connection between brain areas and a measure of correlated gene expression[6,18,21].

Throughout this work, we perform GCEA on biological process GO categories[36] using brain-wide gene-expression data from the AMBA[3] for the mouse brain, and the AHBA[4] for the human cortex. As described in "Methods", enrichment is performed as gene-score resampling[38] using Matlab software that we developed[69]. Given the sensitivity of GCEA results to specific methodological pipelines[48], our aim here is to highlight general issues with applying any GCEA pipeline to spatially embedded

transcriptional atlas data. Thus, while the quantitative results presented here will vary across different GCEA packages and parameters, the statistical biases we characterize will apply to all current GCEA pipelines that assess significance relative to random-gene null models.

**Diverse phenotypes in the literature are enriched for similar GO categories.** We undertook this study after observing a strong similarity in the results of GCEA analyses of spatially embedded data in the literature, despite these analyses involving very different phenotypes, species, measurement modalities, gene-expression processing pipelines, and software implementations of GCEA. To evaluate this consistency, we surveyed 16 mouse GCEA analyses (taken from 8 different studies) and 60 human GCEA analyses (from 23 studies), all involving spatially embedded atlas data from the AMBA and AHBA. Across many very different phenotypes, these studies indeed reported similar significant GO categories, most frequently implicating categories related to metabolic, neuronal, and generic biological and behavioral processes. The most reported GO category in our survey was "chemical synaptic transmission" (GO:000726), which has been reported in 15 human analyses from 10 different studies[21,26,32,72–78] and 4 different mouse analyses from 3 different studies[14,17,79]. These studies collectively implicate genes involved in chemical synaptic transmission in the organization of human resting-state functional connectivity[21], human adolescent cortical shrinkage and myelination[26], and tract-traced structural connectivity in the mouse brain[17]. Some other commonly reported categories include "potassium ion transmembrane transport"[22,26,32,77,80,81], "learning or memory"[15,77,78,82,83], and "electron transport chain"[18,19,23,26,77] (see Supplementary Data 1 for a sorted list of GO categories, annotated by studies). All of these biological processes are sufficiently broad to be plausibly linked to any type of brain-related phenotype, but we aimed to investigate whether the consistency of these findings might instead be driven by common statistical biases in GCEA that favor the selection of some GO categories over others.

**The GO enrichment signature of randomized spatial maps.** We tested for false-positive bias in the application of GCEA to spatial transcriptomic data by characterizing the enrichment results for purely random SBPs (generated by assigning a random number to each brain area independently). We performed GCEA separately for each of 10,000 independent random SBPs, noting which GO categories were significantly related to each SBP (false discovery rate, FDR < 0.05). We then computed the category false-positive rate (CFPR) for each GO category as the proportion of random SBPs for which a statistically significant spatial correlation was identified. This method of computing CFPRs under random SBPs is depicted schematically in Fig. 2A(ii) and labeled "SBP-random". As random SBPs are uninformative, an unbiased GCEA procedure should produce CFPRs consistent with the expected statistical false-positive level, and all GO categories should have similar CFPRs. We estimated this expected CFPR numerically using a "reference" case, shown in Fig. 2A(i), in which each gene's expression data were randomized independently. As this randomization destroys gene–gene coexpression structure in the gene-expression matrix, it allows us to isolate the contribution of nonrandom gene-expression structure to CFPRs through comparison to the "SBP-random" results. Motivated by the strong spatial autocorrelation observed in many real SBPs[6], we also tested whether the use of spatially autocorrelated random phenotypes would affect CFPRs. These spatially autocorrelated SBPs, labeled "SBP-spatial" [Fig. 2A(iii)], were generated using a spatial-lag model[84], with parameters determined from the spatial

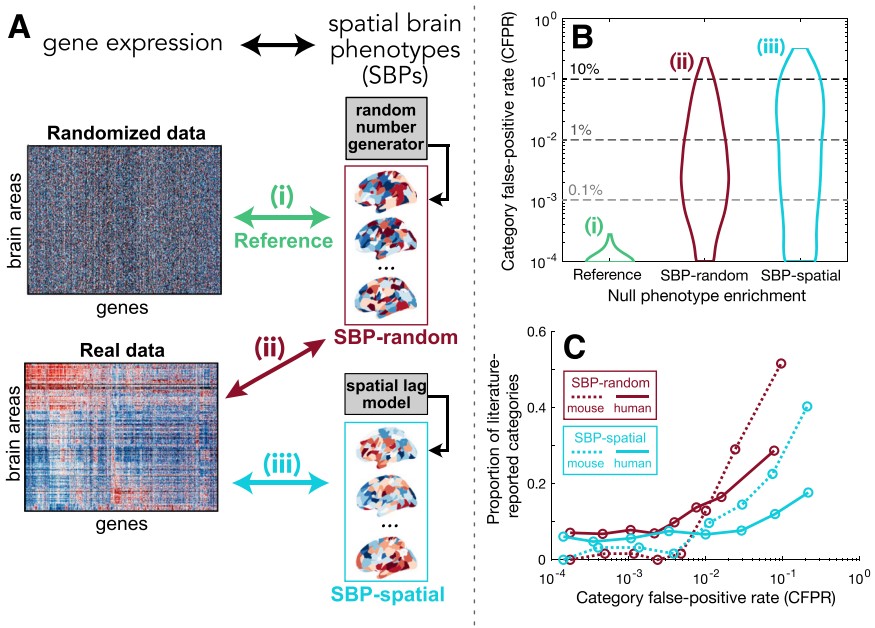

**Fig. 2 Some GO categories have far higher false-positive rates under randomized spatial phenotypes than statistical expectation, and these GO categories are more likely to be reported as significant in published studies. A** A schematic of three GCEA analyses involving correlations between (i) "reference"—an ensemble of random phenotypes and randomized gene-expression data (green), (ii) "SBP-random"—an ensemble of random phenotypes and real gene-expression data (red), and (iii) "SBP-spatial"—an ensemble of spatially autocorrelated phenotypes and real gene-expression data (blue). For the human cortex, examples of spatial maps in each ensemble are plotted; relative to the SBP-random maps that have no spatial correlation structure, the SBP-spatial maps are more likely to have similar values in nearby locations. **B** Distributions of the category false-positive rate (CFPR) across all GO categories are shown as violin plots in mouse. Across an ensemble of 10,000 SBP-random or SBP-spatial maps, the CFPR is computed for each GO category as the proportion of phenotypes for which that GO category was found to be significant. Results are shown for the three analyses depicted in **A**: (i) "reference" (green); (ii) "SBP-random" (red); and (iii) "SBP-spatial" (blue). Note the logarithmic vertical scale (and therefore exclusion of GO categories with CFPR = 0). **C** The proportion of literature-reported GO categories increases with the CFPR estimated from random phenotypes. GO categories were labeled from a literature survey of GCEA analyses using atlas-based transcriptional data in human and mouse (see Supplementary Information for survey details). Across eight equiprobable bins of CFPR (i.e., each bin contains the same number of GO categories), we plot the proportion of all literature-reported GO categories that are contained in that bin. Results are shown for the SBP-random (red) and SBP-spatial (blue) ensembles in mouse (dotted) and human (solid). The position of each bin is shown as the mean of its extremities.

autocorrelation properties of the gene expression data (following Burt et al.[30], see "Methods" for details).

Distributions of nonzero CFPR across GO categories are shown on a logarithmic scale for the three experiments in Fig. 2B performed using mouse expression data (similar quantitative results were obtained in human, cf. Fig. S1). In the "reference" case, consistent with the lack of signal from randomized gene expression and random phenotypes, we observed very low CFPR across our ensemble of 10,000 random SBPs. The vast majority of GO categories (84% in mouse and human) were never judged as statistically significant for any of the 10,000 random maps, and the maximum CFPR was 0.03% in both mouse and human (mean CFPR = 0.002% in both mouse and human).

When repeating the procedure using the same random SBPs, but now with real gene-expression data ["SBP-random", Fig. 2A (ii)], the mean CFPR increased 875-fold in mouse (from 0.002 to 1.5%) and 582-fold in human (from 0.002 to 1.0%). The maximal CFPR jumped dramatically from the reference level of 0.03 to 23% in mouse (to 25% in human). When the ensemble of phenotypes was constrained to exhibit generic spatial autocorrelation structure ("SBP-spatial"), most GO categories decreased their CFPR (60% in mouse and 68% in human). However, some categories exhibited large increases in CFPR, driving an increase in the average CFPR under "SBP-spatial" phenotypes from 1.5 to 2.8% in mouse (maximal CFPR increased from 23 to 37%) and from 1.0 to 2.2% in human (maximal CFPR increased from 25 to 36%). The

differences between the SBP-random and SBP-spatial ensembles will be explored in more detail later.

When applied to spatial transcriptomic data, we thus find major differences in significance testing between controlling false-positive rates relative to randomized genes (conventional GCEA) and controlling false-positive rates relative to randomized phenotypes. Conventional GCEA yields significant false-positive relationships to SBPs made up of random numbers, far beyond statistical expectation, indicating a striking methodological bias. This false-positive bias does not affect all categories equally: some GO categories exhibit reference-consistent CFPR (≤0.03%), while others are judged to be significantly correlated to random spatial maps >20% of the time. Concerningly, the categories with the highest CFPRs in mouse and human are dominated by brain-related function: neurons (e.g., "regulation of synaptic plasticity": 23% in mouse, 10% in human), metabolism (e.g., "respiratory electron transport chain": 19% in mouse and 10% in human), and behavior (e.g., "learning": 20% in mouse and 5% in human), as listed in Table S1 (full list in Supplementary Data 2).

To assess whether GO categories commonly reported as significant in the published literature might be affected by this false-positive bias, we investigated whether these published categories were more likely to have high CFPR. We performed an equiprobable binning of GO categories by their CFPR (placing an equal number of categories in each bin) and then analyzed the proportion of all categories in each bin that had been reported as

statistically significant in the literature. As shown in Fig. 2C, for both ensembles of null phenotypes and in both mouse and human, we find a progressive increase in the frequency of reporting of a GO category with its CFPR. A similar increase was also found when incorporating information about the number of literature studies that have reported a given GO category (Fig. S2). The increase in the reporting of GO categories with their false-positive rate under ensembles of randomized phenotypes suggests that the existing literature is affected by the false-positive biases of conventional GCEA methodology.

**Within-category coexpression drives false-positive bias.** Since randomizing the gene coexpression structure ("reference") dramatically reduces CFPRs, we reasoned that dependencies between genes within a GO category must drive the vast differences in CFPR between categories. We explored how categories of the same size differ in their gene–gene coexpression, using the example of a low-coexpression category ("zymogen activation", CFPR = 0.07%) and a high-coexpression category ("regulation of dendritic spine morphogenesis", CFPR = 13%). As described in Supplementary Section S1, we found that this high-coexpression category is more likely to exhibit high correlations to a random SBP than random genes, because a chance correlation between the random SBP and any single gene is amplified through a similar correlation with many other genes in the category. This mechanism, through which gene–gene dependencies drive deviations from a random-gene null, results in the dramatic differences between false-positive rates estimated from conventional random-gene GCEA and those estimated relative to ensembles of random phenotypes for high-coexpression GO categories.

To test this explanation across all GO categories, we defined a simple measure of within-category coexpression, $\langle r \rangle$, as the mean coexpression across all pairs of genes in a GO category. As shown in Fig. 3A, $\langle r \rangle$ and CFPR (using the SBP-random ensemble) are positively correlated in both mouse and human. Consistent with their coordinated function in the brain, genes associated with metabolic and neuronal functioning in both mouse and human, display characteristic expression patterns resulting in high within-category coexpression, $\langle r \rangle$. For example, among the GO categories with the highest $\langle r \rangle$ are "regulation of short-term neuronal synaptic plasticity" ($\langle r \rangle_{\text{mouse}} = 0.38$, $\langle r \rangle_{\text{human}} = 0.29$) and "ATP synthesis coupled proton transport" ($\langle r \rangle_{\text{mouse}} = 0.34$, $\langle r \rangle_{\text{human}} = 0.29$; see Supplementary Data 3 for a full list).

Our results demonstrate that within-category coexpression plays a key role in driving false-positive bias when applying GCEA to transcriptomic data. And because brain-related categories exhibit high within-category coexpression, they are most prone to this effect, thus driving deceptively sensible brain-related enrichment results (evidenced in published GCEAs across diverse brain phenotypes, Fig. 2C).

**The role of spatial autocorrelation.** We next turn to the role of spatial autocorrelation in SBPs by investigating the SBP-spatial ensemble of spatially autocorrelated phenotypes. We found that adding the constraint of spatial autocorrelation (i.e., SBP-random → SBP-spatial) can be considered a perturbation on a category's CFPR under SBP-random, depending on the spatial autocorrelation structure of genes in that category (see Supplementary Section S1). For example, if the genes in a category exhibit similar spatial autocorrelation structure to that of the phenotype (here, the SBP-spatial ensemble), then that category's CFPR can be substantially inflated.

To demonstrate this, we computed a simple measure of spatial autocorrelation for each GO category, $R^2_{\text{exp}}$, that measures the exponential distance dependence of each category's correlated gene expression (see "Methods"). As shown in Fig. 3B, categories containing genes with stronger spatial autocorrelation scores, $R^2_{\text{exp}}$, are more likely to increase their CFPR under spatially autocorrelated phenotypes (SBP-spatial) relative to random phenotypes (SBP-random). This effect is strongest for categories with an autocorrelation length scale close to that of the SBP-spatial ensemble (Fig. S4), confirming the intuition that a high correlation between a GO category and phenotype is more likely for categories of genes with similar spatial autocorrelation properties as the phenotype. As this effect is not taken into account by conventional GCEA, it can drive increased CFPR, in addition to the effects of gene–gene coexpression.

**Ensemble-based null models for spatial expression data.** The above results reveal clear biases in the application of conventional gene category enrichment methods to transcriptional atlas data, in which controlling the false-positive rate under random genes induces a strong false-positive bias under randomized phenotypes. This bias is strongest in categories of genes that exhibit high within-category coexpression and similar spatial autocorrelation properties as the phenotype of interest. Since brain-related GO categories are characterized by high within-category

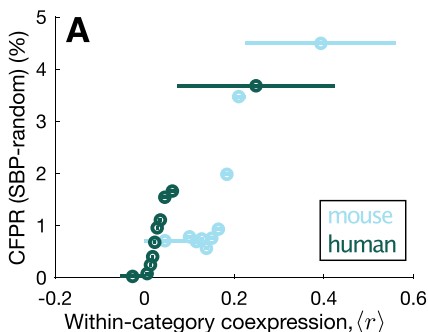
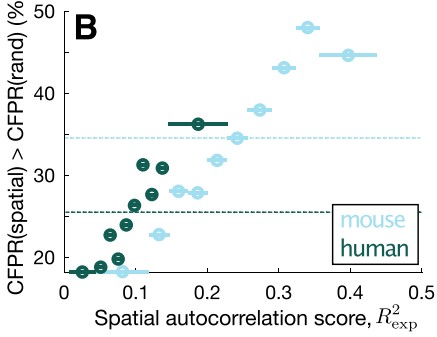

**Fig. 3 Category false-positive significance rates (CFPRs) vary with within-category gene–gene coexpression and spatial autocorrelation in mouse and human. A** CFPR (%) computed from the SBP-rand ensemble of random spatial maps increases with a measure of mean within-category coexpression, $\langle r \rangle$, across ten equiprobable bins in mouse (blue) and human (green). The extent of each bin is displayed as a horizontal line. **B** The percentage of GO categories that exhibited an increase in CFPR when using the SBP-spatial ensemble relative to the SBP-random ensemble, across ten equiprobable bins of the spatial autocorrelation score, $R^2_{\text{exp}}$. This score captures the goodness of fit of each GO category's correlated gene expression to an exponential function with distance. More spatially autocorrelated GO categories are more likely to exhibit an increase in CFPR for spatially autocorrelated phenotypes (the SBP-spatial ensemble). The average value across all GO categories is shown as a horizontal dotted line.

**Fig. 4 The statistical significance of a GO category can be quantified relative to conventional random-gene nulls, or ensemble-based null models introduced here.** For a given spatial brain phenotype (SBP) of interest, we depict the process through which null samples are generated for estimating statistical significance for GCEA across three different null models. **A** The conventional random-gene null tests whether the observed result is more extreme than if genes were assigned to GO categories at random (similar to the illustrated shuffling of gene identities). As this destroys within-category gene–gene correlation structure, it leads to high category false-positive rates for random phenotypes. An alternative is to compute null distributions for each category based on an ensemble of null phenotypes. **B** The SBP-random null tests whether the observed result is more extreme than if the phenotype of interest was a random spatial map. **C** The SBP-spatial null tests whether the observed result is more extreme than if the phenotype of interest was a random spatially autocorrelated map.

coexpression and strong spatial autocorrelation, this false-positive bias misleadingly favors brain-related GO categories. These methodological issues highlight the need for new statistical methods to enable valid inference and interpretation of GCEA for spatially embedded transcriptional data.

In introducing new null models, we imagine a researcher who wishes to investigate the transcriptional correlates of their SBP, asking: "Which GO categories contain genes that are significantly correlated to my phenotype?" The conventional GCEA methodology evaluates the significance of a GO category against a null of random genes, depicted in Fig. 4A (cf. Fig. 1B). This generates null samples by randomizing the annotations of genes to GO categories, and asks the question of each GO category: "Are genes in this category more strongly correlated to my phenotype than a random set of genes?" Through the mechanisms described above, this randomization of gene-to-category annotations destroys within-category gene–gene coexpression structure, thereby driving high CFPRs.

To overcome this bias, we introduce a new way of performing GCEA that, instead of estimating statistical significance relative to randomized gene-to-category assignments, estimates the significance of a given SBP relative to ensembles of randomized SBPs. Two such sets of randomized phenotypes are the SBP-random and SBP-spatial ensembles defined above (Fig. 2A). By estimating each GO category's null distribution relative to an ensemble of randomized phenotypes, ensemble-based nulls test whether a given SBP is significantly correlated to genes in a GO category beyond what would be expected from the null phenotypes. For example, in testing a measured SBP relative to the SBP-random ensemble, the researcher can ask the question of each GO category: "Are genes in this GO category more correlated to my phenotype than they would be to a random phenotype?" (Fig. 4B). Ensemble-based nulls allow for additional constraints (e.g., spatial autocorrelation) to be incorporated straightforwardly. For example, testing against the SBP-spatial ensemble allows the researcher to ask: "Are genes in this GO category more correlated to my phenotype than they would be to a random spatially autocorrelated phenotype?", shown in Fig. 4C. While here we use the generic SBP-spatial ensemble, the spatial autocorrelation properties of the ensemble could be fitted to match the phenotype of interest, to ensure that the spatially autocorrelated ensemble exhibits a similar spatial autocorrelation structure as the phenotype of interest, for which a range of existing methods have been developed[85]. Importantly, ensemble-based nulls preserve the properties of the transcriptomic data (e.g., within-category coexpression and spatial autocorrelation properties of

GO categories) that can inflate CFPRs when performing GCEA using conventional gene randomization.

**Case studies in spatial brain phenotype enrichment.** In this section, we present some case studies to demonstrate how GCEA results can change when assessing significance relative to different null models. We tested two types of SBPs for their GO-category enrichment: (i) regional connectivity metrics, degree, $k$ (defined as the number of connections attached to each region), and betweenness, $B$ (defined as the number of shortest paths on the network passing through each region), derived from binary structural connectomes in mouse and human; and (ii) cell-density maps of parvalbumin (PV)+, somatostatin (SST)+, and vasoactive intestinal peptide (VIP)+ interneurons from mouse cell-mapping experiments[86], and estimated density maps of oligodendrocytes, astrocytes, glia, microglia, neurons, and excitatory and inhibitory neurons[87]. We aimed to understand how the enrichment signatures of these phenotypes differ across the null models introduced above. Full results tables for all GCEA analyses are provided in the data repository accompanying this article[88] (see Supplementary Information).

The conventional (random-gene) null yielded significantly enriched GO categories for 9 of the 14 GCEA analyses performed on mouse (12) and human (2) cortex; these analyses are plotted in Fig. 5A. However, when assessing significance relative to random phenotypes (SBP-rand), only inhibitory cell density (mouse cortex) exhibited significant GO enrichment. When assessing significance relative to random spatially autocorrelated maps (SBP-spatial), no phenotypes were significantly enriched for any GO categories. To illustrate how substantially estimated $p$ values can differ across these three null approaches, we plotted results for oligodendrocyte cell densities across the mouse cortex in Fig. 5B. For example, "aerobic respiration" (GO:0009060) has uncorrected $p$ value estimates of $5 \times 10^{-7}$ (random-gene null), 0.01 (SBP-random), and 0.06 (SBP-spatial). We therefore find that GCEA significance can change markedly across null models and, in most cases investigated here, GO categories considered significantly correlated to a phenotype (relative to random-gene nulls) are in fact statistically consistent with expectation from random phenotypes. Many of the GO categories that are assessed as being significant under the conventional random-gene null model are nonneuronal and hard to feasibly interpret, e.g., "keratinization" correlates with degree, $k$, in human cortex ($q_{FDR} = 1 \times 10^{-7}$), "regulation of kidney development" correlates with neuron density in mouse cortex ($q_{FDR} = 0.02$), and

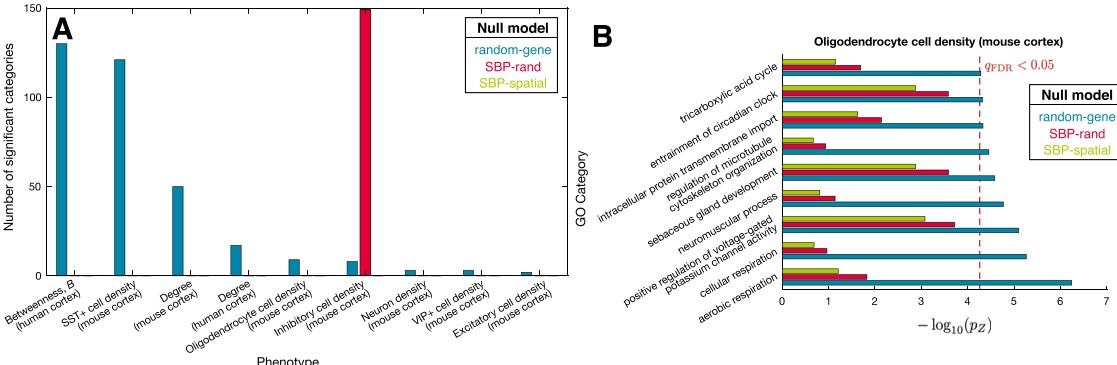

**Fig. 5 GO category enrichment results depend strongly on the null ensemble. A** Across the range of structural connectome nodal metrics (mouse and human) and cell density phenotypes (mouse), we show those nine phenotypes that individually exhibited categories with significant enrichment according to at least one of the null models. In all but one case, enrichment under the random-gene null is not significant under either of the random-phenotype nulls. **B** Picking an example enrichment analysis—oligodendrocyte cell density, which has nine significant categories under the random-gene null—we plot the variation in estimated $p$ values (uncorrected) across the three null models (estimated from a Gaussian fit to the null distribution as $p_Z$). The corrected significance threshold, $q_{FDR} = 0.05$, for the random-gene null is shown as a dashed red line; bars to the right of this line are considered significant at a false discovery rate of 0.05.

"response to pheremone" correlates with VIP+ neuron density in mouse cortex ($q_{FDR} = 0.002$). Thus, the randomized phenotype nulls can eliminate the statistical significance of physiologically inappropriate and spurious GO categories observed under the conventional random-gene null model.

While $p$ values are mostly higher under the phenotype-based nulls, this is not true for inhibitory cell density across mouse cortex[87]. For this phenotype, the random-phenotype null (SBP-rand) flagged more significant categories (149) than the conventional random-gene null (8). That is, of the GO categories that were significantly more correlated to inhibitory cell density than to a random cortical phenotype, most were not more significantly correlated than would be expected from a random set of genes. This discrepancy occurs when an SBP, like inhibitory cell density, aligns strongly with a dominant gene-expression gradient, such that a random gene is far more likely to be correlated to that SBP than to a random SBP. This drives an offset in the random-gene null and increases estimated $p$ values. This effect is important given the dominance of low-dimensional gene-expression gradients in mouse and human cortex[30,89]. Indeed, when expanding our mouse cortical analyses (38 regions) to the whole mouse brain (213 regions), we found a similar mechanism at play, in which the strong cortex–non-cortex expression difference (exhibited by the majority of genes: 73%, 14,137/19,417, $q_{FDR} < 0.05$, Wilcoxon rank-sum test), dominates the enrichment signature of any phenotype that is differentially distributed between cortical and non-cortical areas. As the GO categories with the strongest cortical expression specificity are related to neuronal, metabolic, and behavioral function, GCEA results are correspondingly related to these plausible brain-related functions. This represents another potential source of misinterpretation of applying GCEA to transcriptional atlas data—it is easy to mistakenly assign specificity of enriched GO categories to a given phenotype (e.g., connectivity degree, which is higher in the isocortex), when this enrichment signature is actually driven by a nonspecific expression difference between cortical and non-cortical areas.

Our results reveal that the significance of many phenotypes in GCEA strongly depends on the null model, with most significant findings (relative to random genes) being consistent with statistical expectation (relative to random phenotypes). In the presence of dominant transcriptional gradients, the reverse can also occur: GO categories that are not surprising relative to random-gene annotations (e.g., when the phenotype follows the gradient) can be more surprising relative to random phenotypes. In these cases, a null model matching spatial autocorrelation may offer more interpretable inference. Overall, these results highlight the need for proper comparison and careful consideration of the spatial embedding when applying GCEA to spatial transcriptomic datasets.

## Discussion
The ability to perform GCEA to identify groups of genes putatively related to spatially varying neural phenotypes has been facilitated by open neuroinformatics tools for accessing genome- and brain-wide transcriptional atlases in mouse and human[3–5,90], and accessible software tools for performing GCEA[38–42,46,47]. While previous work has emphasized a general need for care when applying GCEA—including substantial inter-software methodological variance, the reliability of gene-to-category annotations, and ongoing statistical improvements[48–50]—here we present a detailed analysis of methodological issues specifically associated with applications involving spatial transcriptomic data: a new and growing application of GCEA. When assessing the significance of a GO category with respect to spatial correlations of its member genes to an independent phenotype, we show that it makes a substantial difference whether the null model is defined relative to randomized genes (as is conventional) or randomized phenotypes. While both gene- and phenotype-randomization have been proposed previously in the literature[70,71], to our knowledge, only random-gene nulls have been used in the context of transcriptional atlas data, and no previous study has investigated the impact of this choice. We show that the conventional random-gene null judges some GO categories as significantly correlated with a surprisingly high proportion of randomized spatial phenotypes. GO categories with the highest CFPRs have high gene–gene coexpression, and CFPR is further increased when GO categories have a similar spatial autocorrelation structure to the phenotype of interest. Concerningly, high-CFPR categories are more likely to be reported as significant in the literature, suggesting that many published reports may be consistent with the expectations of randomized phenotypes. The flexible new ensemble-based framework that we introduce for generating GO-category null distributions will enable researchers to more accurately interpret the results of applying GCEA to spatially embedded transcriptional data and brain phenotypes.

Applying conventional GCEA to an ensemble of randomized phenotypes led to high rates of estimated significance: relative to the reference level, CFPRs increased by an average of 875-fold in mouse and 582-fold in human. GO categories varied widely in their CFPRs, with some brain-related categories exhibiting CFPRs over 20% (relative to a maximum CFPR of 0.03% in the reference case). We show that this false-positive bias is primarily driven by within-category gene–gene coexpression: a generic property of the expression atlas, not of the specific SBP being analyzed. Categories of genes involved in brain function tend to exhibit spatially coordinated expression patterns (compared to the less correlated expression patterns of non-brain genes), yielding high gene–gene coexpression and thus high CFPR. This leads to the dangerous consequence that applying conventional GCEA to transcriptional atlas data can misleadingly yield significant brain-related GO categories that are feasibly connected to the neural phenotype of interest. For example, GO categories related to oxidative metabolism have been linked to hub connectivity[18,23], a result that can be plausibly interpreted in light of the known metabolic expense of brain network hubs inferred from other modalities[91–93]. While results of conventional GCEA reported the literature are not necessarily spurious, we must take care to validate these results against alternative nulls, like the ensemble-based approach introduced here.

Ensemble-based hypothesis testing for GCEA is a flexible framework with which to perform valid inference of the enrichment of spatial phenotypes. In the SBP-random null model, random numbers are assigned to all brain regions, and in the SBP-spatial null model we instead generate an ensemble of phenotypes with a given spatial autocorrelation structure. The framework is flexible to other types of generative null models to test more specific questions. For example, the SBP-spatial ensemble used here takes parameters from the average spatial correlation properties of transcriptional atlas data, but they could instead be fitted to match the spatial autocorrelation properties of the phenotype of interest, e.g., using the *brainSMASH* toolbox[66]. For data that are sparsely sampled spatially, such that the parameters of these models are difficult to estimate, either the SBP-random null model can be used (which accounts for the major bias of gene–gene coexpression), or the null ensemble could be further constrained by, e.g., additional anatomical factors to test more specific hypotheses. For example, a null ensemble of phenotypes could be generated by randomizing the target phenotype separately within specific anatomical divisions—e.g., the ensemble could preserve differences in expression between cortical and non-cortical areas by randomizing cortical areas separately from subcortical areas. This flexibility to test against different null ensembles allows inference, and subsequent interpretation, to be tailored to the specific scientific question at hand. Furthermore, once a null ensemble has been generated, it can be applied not just to the univariate setting explored in this work, but similarly to multivariate analyses such as those that score genes according to partial least squares or canonical correlation analysis.

Ensemble-based nulls are expensive to compute, as GCEA often requires estimating very small $p$ values and new SBPs must be generated for each null sample. To ease the computational burden, here we used a Gaussian approximation for the 40,000 sample null distribution computed for each GO category, allowing us to estimate small $p$ values from the fitted Gaussian to a greater precision than a direct permutation test. This approximation was appropriate here for characterizing the general mechanisms and trends, but for a specific analysis, a full permutation-based procedure (that does not make strong parametric assumptions on the null distribution) would be necessary. We note that while the initial computation of null distributions for each GO category is expensive, it is a one-off computation for

ensembles that are independent of the phenotype being tested (like the SBP-random and SBP-spatial ensembles studied here). That is, once the initial generic null computation has been computed, ensemble-based GCEA can then be applied efficiently for any new phenotype.

Our case studies of a range of spatial phenotypes across mouse and human cortex demonstrate the need for caution in interpreting the results of GCEA applied to transcriptional brain atlas data. We found that estimated $p$ values varied substantially between the different null models, most typically resulting in GO categories that are surprising relative to random gene annotations, but not surprising relative to randomized phenotypes. This difference highlights the important confounding role of variations in within-category gene–gene coexpression. Statistical sensitivity could be improved by reducing the number of comparisons across GO categories by considering a narrower range of brain-related GO categories of a given size, or by filtering on GO categories containing genes that are not spatially coherent. To this latter point, we found that many GO categories are made up of genes that do not show coherent, characteristic spatial patterning across the brain, and are therefore problematic to interpret as a functionally homogeneous "set". We also found that the effect of nonspecific spatial effects, like cortical versus subcortical expression, can dominate GCEA findings for phenotypes that, e.g., differ between cortical and non-cortical areas, giving a false illusion of specificity and a plausible set of brain-related functional gene categories (that overlap strongly with cortical genes). For phenotypes that follow these types of spatial gradients, random genes are more likely to correlate with the phenotype, resulting in $p$ values that can be higher than those obtained relative to random phenotypes. These discrepancies highlight the need for careful, nuanced interpretation of GCEA results with respect to appropriate null models.

Much of the brain's structural organization can be accounted for by its physical embedding, which imposes a spatial autocorrelation structure on many brain phenotypes. Spatial autocorrelation means that samples are not independent, and requires corrections to any resulting statistical inference, as has been noted in neuroimaging applications[30,66,67,94], in other fields like geography[95], and time-series analysis (temporal autocorrelation)[96–98]. The effect of spatial autocorrelation on GCEA is not straightforward: relative to independent random numbers (SBP-random), applying an auto-correlation constraint (SBP-spatial) yielded a large increase for a minority of GO categories, driving an overall increase in average CFPR. We showed that GO categories with a more similar spatial autocorrelation strength and length scale as the SBP-spatial ensemble have the greatest increase in CFPR relative to the SBP-random ensemble (Fig. 3B and Fig. S4). Differences in CFPRs determined by different null ensembles can be understood by viewing each generative null model as defining a probability distribution in the space of all possible SBPs. For categories that have similar spatial autocorrelation properties as the SBP-spatial ensemble, the SBP-spatial ensemble will define a higher probability density around SBPs that are correlated to its genes, yielding CFPR(SBP-spatial) > CFPR(SBP-random). Because here we use an SBP-spatial ensemble with a fixed strength and length scale, only a minority of categories match this quite restricted set of phenotypes better than SBP-random (40% in mouse and 32% in human), but for categories that match closely, CFPR can increase substantially. For example, in mouse, the CFPR of "ionotropic glutamate receptor signaling pathway" increases from 8.8 (SBP-random) to 29.2% (SBP-spatial) and "dopamine receptor signaling pathway" increases from 6.5 to 26.2%. Interestingly, maps with strong spatial auto-correlation, but at a different length scales, can be less correlated to each other than to random maps; this effect could explain the large drops in CFPR under SBP-spatial of some highly spatially

autocorrelated GO categories such as "ATP metabolic process" (CFPR in mouse decreased from 20.1 to 10.7% under the SBP-spatial ensemble, despite a relatively high spatial autocorrelation score, $R^2 = 0.37$). Thus, whether the SBP-spatial ensemble is more or less conservative than the SBP-random ensemble depends on the specific spatial autocorrelation properties of each given GO category and how they relate to the properties of the SBP-spatial ensemble.

Our main aim here was to characterize the mechanisms of false-positive bias in the application of GCEA to transcriptional data, rather than to verify specific quantitative results of GCEA. Future work will be important to demonstrate how robust and accurate the results of GCEA are to methodological parameters of the GCEA analysis, as well as the choices made in processing gene-expression data[5]. Refinements to the GCEA pipeline will also be important, and may include developing inferential methods that account for the hierarchical organization of GO categories[48,52]; pruning the set of categories analyzed to those that are brain relevant; and extending the current work to consider analyses of pairwise inter-regional (e.g., connectivity-based) phenotypes[6,10,18]. We also note that while GO annotations were used here, the same arguments apply to other annotation systems, like the annotation of gene markers to cell types[99–101], where similar cautions should be noted.

Finally, we make some recommendations to improve the transparency and reproducibility of applying GCEA to transcriptomic atlas data. In reporting on GCEA analyses, we note that GO annotations are updated daily, such that GCEA tools can produce different results to identical inputs when run at different times. We thus recommend that researchers share the gene-score or gene-list inputs they used as input to the GCEA pipeline with a detailed methodological description, including all settings used in performing enrichment. This would allow future studies to test for reproducibility through time, as gene annotations, categories, and category structures are updated[49], and would enable testing for robustness to other methodological implementations of GCEA (e.g., from different software packages). In reporting the results of GCEA, we discourage the practice of only listing manually selected GO categories in manuscript text but, rather, full outputs of the enrichment procedure should be provided as supplementary files that include gene scores and raw (and corrected) $p$ values.

The public availability of gene transcriptional atlases provides an opportunity to bridge spatial scales in the brain to understand how variations in molecular function relate to large-scale properties of brain structure and function[6,102]. GCEA plays a crucial methodological role in leveraging gene ontologies to interpret these multi-scale relationships. We describe major statistical issues in applying conventional implementations of GCEA to assess the correlation between SBPs and the expression patterns of functionally annotated categories of genes, highlighting striking differences between nulls obtained from randomizing gene-to-category annotations (as is conventional) versus randomizing phenotypes. The new ensemble-based null models introduced here will allow researchers to better interpret the results of applying GCEA in using spatially resolved molecular maps to address diverse scientific questions. Extensive data tables for all enrichment analyses, code for reproducing all presented results, and a toolbox for performing ensemble-based enrichment accompany this paper.

## Methods

**Mouse data**. We used gene-expression data from the AMBA[3] using the 213-region parcellation of Oh et al.[103]. Data were retrieved using the Allen Software Development Kit alleninstitute.github.io/AllenSDK/. Gene transcription in a given brain area was summarized as the "expression energy" (the mean ISH intensity across voxels of that brain area)[3,104]. Where multiple section datasets were available for

the same gene, they were combined by first $z$-scoring expression data taken across areas, and then computing the expression value of a cortical area as the mean across these $z$-scored section datasets. For each of 19,419 genes, we applied a 50% quality threshold, first on genes (keeping genes with expression data for at least 50% of brain areas), and then on brain areas (keeping brain areas with expression data for at least 50% of genes). This resulted in a region × gene expression matrix of size $213 \times 19\,417$ (whole brain), and $38 \times 19\,417$ (isocortical areas).

For the case study involving structural connectivity degree, we used axonal-connectivity data based on 469 anterograde viral microinjection experiments in C57BL/6J male mice at age P56, obtained from the Allen Mouse Brain Connectivity Atlas[103]. We computed degree, $k$, from binarized ipsilateral axonal connectivity in the right-hemisphere, including edges with $p < 0.05$, as inferred from the whole-brain linear-regression model presented in Oh et al.[103] (see Fulcher and Fornito[18]).

**Human data**. We used gene-expression data from the AHBA that consists of microarray expression measures across 3702 spatially distinct tissue samples collected from six neurotypical postmortem adult brains[4]. Considering that two of the six brains were sampled from both left and right hemispheres, while the other four brains had samples collected only from the left hemisphere, we focused our analyses on the left hemisphere only. The data were processed as outlined in Arnatkevičiūtė et al.[5]. Specifically, (i) probe-to-gene annotations were updated using the Re-Annotator toolbox[105]; (ii) intensity-based filtering was applied in order to exclude probes that do not exceed background noise in >50% of samples; (iii) if more than one probe was available for a gene, a probe with the highest differential stability score[57] was selected; (iv) gene-expression samples from cortical regions were assigned to the regions-of-interest by generating donor-specific gray matter parcellations (180 regions per hemisphere[106]) and assigning samples located within 2 mm of the parcellation voxels; and (v) gene-expression measures within a given brain were normalized first by applying a scaled robust sigmoid normalization[107] for every sample across genes and then for every gene across samples. This allowed us to evaluate the relative expression of each gene across regions, while controlling for donor-specific differences in gene expression. Normalized expression measures in samples assigned to the same region were averaged within each donor brain and then aggregated into a $180 \times 15\,744$ region × gene matrix containing expression measures for 15,744 genes across 180 cortical regions in the left cortex. Three of the 180 regions had no gene expression samples assigned; all analyses shown here are of the remaining 177 regions.

Structural connectivity was estimated based on the minimally processed diffusion weighted imaging and structural data from the Human Connectome Project[108] for 972 participants (age$_{mean}$ = 28.7 ± 3.7, 522 females)[109,110]. Data were acquired on a customized Siemens 3T "Connectome Skyra" scanner (Washington University in St Louis, MO, USA) using a multi-shell protocol for the DWI: 1.25 mm³ isotropic voxels, repetition time (TR) = 5.52 s, echo time (TE) = 89.5 ms, field-of-view (FOV) of $210 \times 180$ mm, 270 directions with $b = 1000, 2000, 3000$ s/mm² (90 per $b$ value), and 18 $b = 0$ volumes. Structural T1-weighted data were collected using 0.7 mm³ isotropic voxels, TR = 2.4 s, TE = 2.14 ms, FOV of $224 \times 224$ mm. The full details regarding data acquisition can be found elsewhere[109]. For each individual network, nodes were defined using a recently-developed, data-driven group average parcellation of the cortex into 360 regions (180 per hemisphere)[106] using Freesurfer-extracted surfaces and subsequently projected to volumetric space.

Processing of the DWI data were performed using the MRtrix3 (ref. [111]) and FMRIB Software Library[112]. Tractography was performed in each participant's T1 space using second order integration over fiber orientation distributions (iFOD2)—a probabilistic algorithm that improves the quality of tract reconstruction in the presence of crossing fibers and high degree of fiber curvature[113]. To further improve the biological accuracy of the structural networks we also applied anatomically constrained tractography, which delineates the brain into different tissue types (cortical gray matter, subcortical gray matter, white matter, and cerebrospinal fluid). This information is then used to ensure that streamlines are beginning, traversing, and terminating in anatomically plausible locations[114]. Tissue types were determined using FSL software[112]. A total of $10^7$ streamlines were generated using a dynamic seeding approach. By evaluating the relative difference between the estimated and current reconstruction fiber density, it preferentially initiates seeding from areas of insufficient density[115].

The resulting tractogram was then combined with the cortical parcellation for each subject assigning streamline termination points to the closest region within 5 mm radius. Connection weights were quantified using streamline count (number of streamlines connecting two regions). Connectomes derived from probabilistic tractography algorithms are often thresholded due to the high probability of false-positive connections[116,117], therefore a single group-average connectome was aggregated by selecting connections that are present in at least 30% of subjects and retaining the strongest edges (based on the median streamline count across subjects) to achieve a connectome density of 20%.

**Spatial phenotypes**. We computed degree, $k$, and betweenness, $B$, of each node from the binary connectomes described above. Node betweenness was computed using the Brain Connectivity Toolbox[118]. When estimating these quantities in the mouse cortex, only cortico-cortical connectivity was used. Interneuron subtype densities were measured by $q$Brain, quantitative whole-brain mapping of

distributions of fluorescently labeled neural cell types[86]. For each brain region, we took the mean cell density (across ten repeat experiments) for each of PV-containing, SST-containing, and VIP-containing cells, using data provided from the authors. Cell densities were taken from the atlas described in Erö et al.[87], which inferred cell densities computationally from Nissl stains, and used genetic cell-type markers from AMBA data to estimate cell densities of astrocytes, oligodendrocytes, glia, microglia, neurons, and excitatory and inhibitory neurons.

**Gene category enrichment analysis**. We performed GCEA using gene-score resampling, where continuous scores are assigned to each gene, aggregated as the mean across individual GO categories, and then compared to a null distribution of category-level scores to perform statistical inference on GO categories[36]. We first describe our implementation of conventional GCEA using gene-score resampling against random-gene nulls (ensemble-based nulls are described later). We performed GCEA using Matlab-based software that we developed[69]. We used GO term hierarchy and annotation files were downloaded from GO on 17th April 2019. We first matched annotations to GO terms (excluding NOT and ND annotations[48]). To match genes listed in the GO annotation file (MGI identifiers) to expression data (NCBI identifiers), we used MouseMine[119]. In human, identification was made directly from the gene symbol. Annotations to genes that could not be mapped in this way were ignored. Direct gene-to-category annotations were propagated up to parents within the GO-term hierarchy by taking the `is_a` keyword to indicate parent–child relationships. This process yielded a fully propagated set of GO categories and their gene annotations. We restricted our analyses to GO categories related to biological processes with between 10 and 200 annotations. We performed enrichment across all GO categories that met these criteria.

Null distributions for a GO category of a given size, $s$, were generated through 40,000 random samples of $s$ gene scores. Achieving stable $p$ value estimates from the empirical null distribution generated through permutation testing can require millions of null samples due to the small $p$ values involved in GCEA. As null distributions of category scores tend to be approximately Gaussian, we chose to estimate $p$ values from a Gaussian distribution fitted to a given set of null samples. That is, $p$ values were estimated numerically after fitting a Gaussian distribution to the estimated null distribution. We note that null distributions are not exactly Gaussian, and that this approximation leads to a systematic underestimate of $p$ values relative to a full, permutation-based approach. As our main focus was to elucidate the mechanisms of bias of GCEA (rather than reporting precise $p$ values), we used this approximation throughout for computational efficiency. Correction for multiple-hypothesis testing was achieved as a FDR[120], denoted as $q_{FDR}$ and computed using the `mafdr` function in Matlab.

**Generating spatially autocorrelated maps**. We generated SBP-spatial ensembles of spatially autocorrelated SBPs using the spatial-lag model[30,84]. Code was written in Matlab, adapted from the Murray Lab's `surrogates` package by Joshua Burt, available at https://bitbucket.org/murraylab/surrogates/src/master/[30]. Note that the *BrainSMASH* package has since been developed: https://github.com/murraylab/brainsmash[66]. In the spatial-lag model, for a Euclidean distance matrix constructed for all areas, two parameters determine the generated maps: a parameter controlling the strength of the spatial autocorrelation (relative to noise), $\rho$, and the characteristic spatial scale, $d_0$. We set $\rho = 0.8$ and estimated $d_0$ from an exponential fit to (Pearson) correlated gene expression (across all genes) as a function of distance[18], yielding $d_0 = 1.46$ mm in mouse brain, $d_0 = 1.84$ mm in mouse cortex, and $d_0 = 102.2$ mm in human cortex.

**The within-category coexpression score, $\langle r \rangle$**. To investigate the role of within-category coexpression in driving false-positive significance, we computed a simple measure of within-category coexpression, $\langle r \rangle$. For each GO category, we computed the gene × gene coexpression matrix, $C$, for all genes in that category as Spearman correlation coefficients between the spatial expression pattern of each pair of genes. The within-class coexpression metric, $\langle r \rangle$, was then computed as the mean of the unique (upper diagonal) entries of $C$.

**The spatial autocorrelation score, $R^2_{exp}$**. To investigate the role of spatially autocorrelated SBPs in driving differential significance between GO categories, we computed a spatial autocorrelation score, $R^2_{exp}$, for each GO category. For each category, we computed a correlated gene expression (CGE) value for each pair of brain regions as a Pearson correlation between the expression values of genes in that category. We then fitted a three-parameter exponential function to the variation of CGE with distance, $d$, as $CGE(d) = A \exp(-d/\lambda) + B$. The goodness of this fit, measured as $R^2_{exp}$, provides a measure of how well a given category of genes exhibits a decaying distance-dependent expression similarity. $R^2_{exp}$ will be high for strongly spatially autocorrelated categories of genes. For spatially autocorrelated GO categories, the length scale, $\lambda$, provides a measure of the autocorrelation length scale (used in Fig. S4). Note that we obtained similar results when computing a Spearman correlation between $d$ and CGE (as the statistic $-\rho$), which does not assume the exponential functional form. Values of these fitted parameters for each GO category are in Supplementary Data 4.

**Reporting summary**. Further information on research design is available in the Nature Research Reporting Summary linked to this article.

## Data availability

All raw data used in this study were obtained from publicly available sources. Raw data and our processed versions of them are available from two Zenodo data repositories accompanying this article[88,121]. Instructions and code for retrieving these data from their sources, and processing them, are in the code repositories described in the "Code availability" section below. A summary of available data is provided here; detailed descriptions of processing pipelines are in the "Methods" section above. The first set of data files used are required to perform general GCEA analyses (including ensemble-based enrichment), including information about the GO hierarchy and gene-to-category annotations[121]. This repository includes GO term hierarchy (the `term` and `term2term` tables from a the `GODaily` mySQL database dump), and GO term annotation files for *Mus musculus* (`mgi.gaf`) and *Homo sapiens* (`goa_human.gaf`), which were downloaded on 17th April 2019. Data for mapping MGI gene identifiers to NCBI Entrez gene identifiers were obtained from *MouseMine*[119], yielding the data file `ALL_MGI_ID_NCBI.csv` (generated September 2017). The second set of data are required to reproduce all analyses produced here, including the transcriptional atlas data, the spatial brain phenotypes used for the case studies, and output data files from our analyses[88]. This includes human gene-expression data from the Allen Human Brain Atlas (AHBA)[4] and processed according to Arnatkevičiūtė et al.[5]; mouse gene-expression data from the Allen Mouse Brain Atlas (AMBA)[3] and processed according to Fulcher et al.[89]; human connectome data from the Human Connectome Project[108]; mouse connectome data from the Allen Mouse Connectivity Atlas[103]; and mouse cell-type maps estimated experimentally Kim et al.[86] and computationally Erö et al.[87].

## Code availability

A combination of Matlab 2020a and python were used for data retrieval and processing, and Matlab was used for all analyses. Fully documented code for reproducing all analyses presented here is provided with this article, https://github.com/benfulcher/GCEA_FalsePositives[122]. A toolbox for performing the conventional (random-gene null) and ensemble-based GCEA described here is also available, https://github.com/benfulcher/GeneCategoryEnrichmentAnalysis[69]. This repository includes instructions for processing raw GO hierarchy and annotation data files (in the repository's Github wiki).

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

## Acknowledgements

The authors would like to thank Stuart Oldham for assistance in processing human structural connectivity data, and Oliver Cliff for thoughtful discussion and feedback on the manuscript. We thank Colorgorical[123] for assisting with the generation of color palettes. A.F. was supported by the Sylvia and Charles Viertel Charitable Foundation.

## Author contributions

B.D.F. carried out the analysis. B.D.F. and A.A. processed the data. B.D.F., A.A., and A.F. contributed to the writing of the manuscript. A.F. supervised the work.

## Competing interests

The authors declare no competing interests.
