## [Peer Review File · Nature Communications]

Reviewers' Comments:

Reviewer #1:

Remarks to the Author:

In this revised manuscript the authors have presented compelling arguments that gene co-expression and spatial autocorrelation of gene expression is responsible for inflation in the significance and interpretation of gene ontology results in neuroimaging studies. The authors have addressed most of the concerns of the reviews and it is important for researchers to be aware of these biases in interpreting large scale transcriptomic data and their implications. The manuscript has also been made more concise and readable. There are a few issues that need to prevent the manuscript from publication but are worth raising.

1. It is probably worth mentioning in the paper, as I said in my first review that these issues of gradient, co-expression, and auto-correlation have been observed now at the cellular and cell type level and that it is basic characteristic of brain genome biology. Thus, the statistical treatment of these effects is not restricted to large tissue-based assays, such as microarrays, but continue to be seen as the field produces high resolution brain wide single cell datasets. Furthermore, GO analyses of cellular level data has been largely unfruitful as the annotations themselves have all been deduced from tissue based large structural assays for the most part. Undoubtedly this will change moving forward, but the applicability of these methods in the future will still be appropriate.
2. The section on Case studies: Spatial brain phenotype enrichment is better presented now and highlights the value of the approach.
3. Reviewer 3 raises important points. I believe this methodology and consideration of spatial relationships when considering significance is the key idea. The applicability of this to interpretation of GO results is in fact I believe secondary. The use of GO results does supply a means to propose and interpret associations but as reviewer 3 notes, few of these associations are interesting or reflect actual novel biology. The authors may wish to slightly refocus the value of the method toward the computation of the appropriate spatial background for any interpretive result whether it is gene expression alone, differential expression between regions, or a specific GO annotation. Reviewer 3's perspective on this matter is I believe quite correct. One way to accommodate this would be simply to say that we use GO annotation as a metric for identifiability of a result, although the method is more widely applicable.
4. As other reviewers have noted the authors use of GSEA is rather unfortunate as the method of Subramanian et al has really locked in the reference of this prior method which has been rather extensively used. A simple acronym change would alleviate future confusion with the new method. In particular as the core of the result is really the SBP-spatial methodology applied to any interpretative conclusion.

Reviewer #2:

Remarks to the Author:

The authors have comprehensively addressed all my concerns and I recommend publication.

Reviewer #3:

Remarks to the Author:

I liked this manuscript as originally submitted, and my suggestions to improve it have been taken up to my satisfaction. A key improvement was to clarify and strengthen the message that past applications of this type of analysis are likely mostly spurious. I also appreciate that I may have underestimated how popular this type of analysis is becoming. The author's work could help stamp out further growth of inappropriate analyses and especially any tendency for people to take them seriously as real results.

Reviewer #1

In this revised manuscript the authors have presented compelling arguments that gene co-expression and spatial autocorrelation of gene expression is responsible for inflation in the significance and interpretation of gene ontology results in neuroimaging studies. The authors have addressed most of the concerns of the reviews and it is important for researchers to be aware of these biases in interpreting large scale transcriptomic data and their implications. The manuscript has also been made more concise and readable. There are a few issues that need to prevent the manuscript from publication but are worth raising.

We thank the reviewer for their thoughtful reading of our manuscript.

1. It is probably worth mentioning in the paper, as I said in my first review that these issues of gradient, co-expression, and auto-correlation have been observed now at the cellular and cell type level and that it is basic characteristic of brain genome biology. Thus, the statistical treatment of these effects is not restricted to large tissue-based assays, such as microarrays, but continue to be seen as the field produces high resolution brain wide single cell datasets. Furthermore, GO analyses of cellular level data has been largely unfruitful as the annotations themselves have all been deduced from tissue based large structural assays for the most part. Undoubtedly this will change moving forward, but the applicability of these methods in the future will still be appropriate.

We agree that emphasizing the broader applicability of our work to new assays at the cellular level is important. We have flagged this important application in the Introduction:

“But growing applications of GCEA to spatial transcriptional data—at the whole-brain as well as microscopic scale (Edsgård et al., 2018; Sun et al., 2020)—are associated with unique challenges due to the data’s spatial embedding.”

“We focus on whole-brain analyses here but note that the same principles apply to GCEA analyses on any scale.”

Similar edits have been made throughout to emphasize this point that the whole-brain scale is just used here for demonstration. E.g., in Results: *“The SBP is a spatial map of some measurement (e.g., taken across brain areas),”* and in several places rewording *‘transcriptional atlas data’* to the more general *‘spatial transcriptomic data’* (where appropriate).

References:

- Edsgård et al. Identification of spatial expression trends in single-cell gene expression data. *Nature Methods* **15**, 339 (2018).
- Sun et al., Statistical analysis of spatial expression patterns for spatially resolved transcriptomic studies. *Nature Methods* **17**, 193 (2020).

2. The section on Case studies: Spatial brain phenotype enrichment is better presented now and highlights the value of the approach.

3. Reviewer 3 raises important points. I believe this methodology and consideration of spatial relationships when considering significance is the key idea. The applicability of this to interpretation of GO results is in fact I believe secondary. The use of GO results does supply a means to propose and interpret associations but as reviewer 3 notes, few of these associations are interesting or reflect actual novel biology. The authors may wish to slightly refocus the value of the method toward the computation of the appropriate spatial background for any interpretive result whether it is gene expression alone, differential expression between regions, or a specific GO annotation. Reviewer 3's perspective on this matter is I believe quite correct. One way to accommodate this would be simply to say that we use GO annotation as a metric for identifiability of a result, although the method is more widely applicable.

We agree that a better appreciation of spatial statistics, and in particular, that non-independence of samples (e.g., due to spatial autocorrelation), is an important methodological consideration in general. Our focus here is on the issues specific to GO, including the major consistent effect of within-category coexpression. We have modified the manuscript to better emphasize how our work, and that of others, has now developed a range of general tools for valid statistical inference in the presence of spatial dependencies:

Issues related to spatial autocorrelation of brain data have been highlighted in other contexts, with researchers developing methods to better estimate null distributions in the presence of spatial autocorrelation, e.g., using spatial permutation methods like spatial-lag models (Burt et al., 2018; Burt et al., 2020) and spin tests (Alexander Bloch et al., 2018) to test against an ensemble of surrogate spatial maps, or by removing the effect of physical distance through regression (French et al., 2011; Ji et al., 2014; Fakhry et al., 2015a; Fakhry et al., 2015b; Reddy et al., 2018; Betzel et al., 2019; Fulcher et al., 2016; Arnatkeviciute et al., 2019).

And in the Discussion:

Spatial autocorrelation means that samples are not independent, and requires corrections to any resulting statistical inference, as has been noted in neuroimaging applications (Breakspear et al., 2014; Burt et al., 2018; Alexander Bloch et al., 2018; Burt et al., 2020), and other fields like geography (Cardillo et al., 2015) and time-series analysis (Afyouni et al., 2019; James et al., 2019; Cliff et al., 2021).

4. As other reviewers have noted the authors use of GSEA is rather unfortunate as the method of Subramanian et al has really locked in the reference of this prior method which has been rather extensively used. A simple acronym change would alleviate future confusion with the new method. In particular as the core of the result is really the SBP-spatial methodology applied to any interpretative conclusion.

To avoid confusion, we now use the term "Gene Category Enrichment Analysis (GCEA)" to refer to the general suite of any method that attempts to do statistical inference on gene scores

or gene sets at the level of annotated categories. We have changed the terminology throughout, including in an updated title.